# Supramolecular metal-organic frameworks that display high homogeneous and heterogeneous photocatalytic activity for $H_2$ production

Jia Tian[1], Zi-Yue Xu[1], Dan-Wei Zhang[1], Hui Wang[1], Song-Hai Xie[1], Da-Wen Xu[1], Yuan-Hang Ren[1], Hao Wang[1], Yi Liu[2] & Zhan-Ting Li[1]

Self-assembly has a unique presence when it comes to creating complicated, ordered supramolecular architectures from simple components under mild conditions. Here, we describe a self-assembly strategy for the generation of the first homogeneous supramolecular metal-organic framework (SMOF-1) in water at room temperature from a hexaarmed $[Ru(bpy)_3]^{2+}$-based precursor and cucurbit[8]uril (CB[8]). The solution-phase periodicity of this cubic transition metal-cored supramolecular organic framework (MSOF) is confirmed by small-angle X-ray scattering and diffraction experiments, which, as supported by TEM imaging, is commensurate with the periodicity in the solid state. We further demonstrate that SMOF-1 adsorbs anionic Wells−Dawson-type polyoxometalates (WD-POMs) in a one-cage-one-guest manner to give WD-POM@SMOF-1 hybrid assemblies. Upon visible-light (500 nm) irradiation, such hybrids enable fast multi-electron injection from photosensitive $[Ru(bpy)_3]^{2+}$ units to redox-active WD-POM units, leading to efficient hydrogen production in aqueous media and in organic media. The demonstrated strategy opens the door for the development of new classes of liquid-phase and solid-phase ordered porous materials.

[1] Department of Chemistry, Collaborative Innovation Center of Chemistry for Energy Materials (iChEM), Fudan University, Shanghai 200433, China. [2] The Molecular Foundry, Lawrence Berkeley National Laboratory, Berkeley, California 94720, USA. Correspondence and requests for materials should be addressed to H.W. (email: wanghui@fudan.edu.cn) or to Y.L. (email: yliu@lbl.gov) or to Z.-T.L. (email: ztli@fudan.edu.cn).

Metal-organic frameworks (MOFs) are periodic porous architectures that are made by linking inorganic and organic units with strong bonds[1–8]. MOFs have many potential applications in, such as, adsorption and separation[9–12], catalysis[13–15], drug delivery and biomedical imaging[16–19] and optoelectronics[20,21]. Their solid nature, however, can impose limitations such that nearly all function-related studies are performed in a heterogeneous manner. For drug delivery and biomedical imaging, this means the occurring of phase separation in the body, which may lead to further harmful aggregation and/ or biological incompatibility[4]. On the other hand, with a handful of exceptions[22,23], most of the reported MOFs are prepared using hydrothermal and solvothermal techniques that require high temperature and long reaction time. These techniques allow for slow growth of crystals for attaining periodicity, but may restrict simultaneous introduction of many less stable functional groups[24]. Self-assembly has been demonstrated to be a robust and mild tool for the generation of advanced and complicated architectures[25,26], such as molecular capsules[27,28], interlocked superstructures[29] and supramolecular polymers and gels[30–37]. With this strategy, crystalline supramolecular and hydrogen-bonded organic frameworks have been constructed as porous materials for gas adsorption[38,39]. MOFs that integrate heterogeneous supramolecular host-guest chemistry by introducing macrocyclic hosts to the struts have also been reported[40]. Recently, we and other groups have illustrated that homogeneous supramolecular organic frameworks can be realized in water in two-dimensional (2D) and three-dimensional (3D) spaces[41–45], which offer promise for the creation of new water-soluble frameworks with the feature of typical MOFs under mild conditions.

Herein, we report that self-assembly can be applied to fabricate the first water-soluble supramolecular MOF, SMOF-1, from a hexaarmed [Ru(bpy)$_3$]$^{2+}$-based precursor and cucurbit[8]uril (CB[8]). We reveal that SMOF-1 maintains its periodicity in both water and the solid state. We further illustrate that this cubic supramolecular MOF can adsorb anionic Wells–Dawson-type polyoxometalate (WD-POM) clusters in a one-cage-one-guest manner and, on 500 nm visible light excitation, the framework facilitates fast multi-electron injection from the [Ru(bpy)$_3$]$^{2+}$ units to the encapsulated WD-POM anions, leading to remarkably efficient hydrogen production in both aqueous and organic medium.

## Results

**The design and synthesis of target and control compounds.** Previous studies established that CB[8] remarkably stabilizes the homodimerization of the 4-phenylpyridin-1-ium (PhPy) unit through hydrophobically driven encapsulation in water[46,47]. We thus prepared hexaarmed [Ru(bpy)$_3$]$^{2+}$ complex **1** from the reaction between **2** and Ru(DMSO)$_4$Cl$_2$ (Fig. 1). The [Ru(bpy)$_3$]$^{2+}$ complex core was soluble and highly stable in water, while its rigid octahedral nature was expected to facilitate the formation of a cubic periodic framework through the 2:1 encapsulation of the appended PhPy units by CB[8].

**$^1$H NMR, DLS and mass spectrometric studies.** To study the binding motif between the PhPy units of **1** and controls **2** and **3** (Fig. 1) and CB[8], the $^1$H NMR spectra of the three mixed solutions in D$_2$O were first recorded by keeping (PhPy)/ (CB[8]) = 2 (Supplementary Figs 1–3). In all the spectra, the signals of the PhPy unit of compounds **1**–**3** shifted upfield considerably, indicating that this aromatic unit was encapsulated by CB[8]. Job plots obtained by fluorescence or absorption experiments confirmed that, for all three mixtures, this encapsulation occurred in a 2:1 stoichiometry (Supplementary Figs 4–6). The $^1$H NMR spectrum of the 2:1 solution of **3** (4.0 mM) and CB[8]

**Figure 1 | Compounds used in this study.** The structures of compounds **1**–**3** and CB[8].

displayed one set of signals, also suggesting the formation of a 2:1 complex (**CP-a**, Supplementary Fig. 7). This complexation motif was further confirmed by the electrospray ionization mass spectrometry of the mixture solution, which exhibited an ion peak at $m/z = 1{,}030.8577$ corresponding to $[3_2 + \text{CB}[8] - 2\text{Cl}]^{2+}$ (calculated value: 1,030.8577; Supplementary Fig. 8). The $^1$H NMR spectrum of the 1:1 solution of ditopic **2** (2.0 mM) and CB[8] also exhibited one set of signals. Addition of more CB[8] did not cause the signals of the aromatic protons of **2** to shift or the appearance of new signals. We tentatively proposed that the two compounds self-assembled into the macrocyclic $3 + 3$ complex $2_3 \bullet \text{CB}[8]_3$ (**CP-b**, Supplementary Fig. 7), on account of the size match between the model of this $3 + 3$ complex and the hydrodynamic diameter ($D_\text{H}$) obtained from dynamic light scattering (DLS) experiment (*vide infra*). The $^1$H NMR spectrum of the 1:3 solution of **1** (1.0 mM) and CB[8] displayed broad signals, which remained unchanged at higher temperature or on the addition of more CB[8], implying that a more complicated complex (**CP-c**) was produced. In CD$_3$CO$_2$Na-buffered D$_2$O (50 mM, pD = 4.74), the apparent association constant ($K_\text{a}$) of the 2:1 complex formed between the PhPy unit of compounds **1**–**3** and CB[8] was determined to be $3.1 \times 10^{14}$, $6.9 \times 10^{12}$ and $5.4 \times 10^{10}$, respectively[48]. From **3** to **2** and then to **1**, the $K_\text{a}$ value increased significantly. This result suggested a positive cooperativity for the cases of **3** and **2**, thus excluding the formation of linear disordered supramolecular polymers by **2** and, for **1**, supporting the formation of highly ordered supramolecular entities. 2D diffusion-ordered spectroscopic $^1$H NMR experiments were also performed for the three mixture solutions in D$_2$O by keeping [PhPy]/[CB[8]] = 2 and [PhPy] = 6.0 mM. For all the mixtures, the signals of the two components exhibited a similar diffusion coefficient ($D$) ($1.9 \times 10^{-10}$, $1.3 \times 10^{-10}$ and $1.2 \times 10^{-11}\,\text{m}^2\,\text{s}^{-1}$ (Supplementary Figs 9–11). The value of **CP-c** was considerably lower than that of **CP-a** or **CP-b**, supporting that this mixture formed larger supramolecular entities. DLS experiments were also performed for the three mixtures in water. With [PhPy]/[CB[8]] = 2 and [PhPy] = 12.0 mM, the three mixtures gave rise to a $D_\text{H}$ value of 2.70, 3.62 or 164 nm, respectively (Supplementary Fig. 12). The fact that the $D_\text{H}$ of **CP-c** was >45 and 60 times larger than that of **CP-b** and **CP-a**, respectively, clearly confirmed that **1** and CB[8] self-assembled into large supramolecular entities. Indeed the $D_\text{H}$ of **CP-c** showed a monotonic rise in response to the increase of [PhPy] (Supplementary Fig. 13). At [PhPy] = 36.0 mM, the $D_\text{H}$ of **CP-c** increased to 250 nm and at higher concentrations, the complex began to precipitate.

**Characterization of 3D cubic framework.** Compound **1** has a rigid octahedral geometry. In an ideal situation, its binding with CB[8] would produce 3D cubic supramolecular entities with [Ru(bpy)$_3$]$^{2+}$ as the vertex of the net and CB[8] encapsulating two PhPy units of neighbouring molecules of **1**. The crystal structure of such a 3D network was then simulated using Materials Studio 7.0 (Accelrys Materials Studio Release Notes, Release 7.0, Accelrys Software Inc., San Diego, USA) and shown in Fig. 2. Although **1** is a racemate of two dynamically stable enantiomers as a result of the bidentate nature of the bpy ligand, we chose to simulate the 3D structure from only the Λ enantiomer, because parameters that define the periodicity of the two resulting frameworks should be identical.

To verify the possible periodicity of the new 3D supramolecular network formed in the aqueous solution of **CP-c** ([**1**] = 3.0 mM), synchrotron small-angle X-ray scattering (SAXS) experiment was performed, which gave a strong peak related to

the d-space centred at around 3.1 nm (Fig. 3a). The value matched well with the {100} spacing (3.0 nm) of the modelled network. This peak persisted even at a low concentration of 0.6 mM (for **1**; Supplementary Fig. 14). The broadness of the peak may be due to the dynamic nature of the supramolecular framework in solution. The solution-phase synchrotron X-ray diffraction profile displayed two broad, but discernible peaks around 3.0 and 2.1 nm (Fig. 3b), which can be assigned to the spacing of the {100} and {110} (2.1 nm) faces. Both experiments supported that **CP-c** formed soluble periodic supramolecular MOF (SMOF-1) in water. Fluorescent experiments showed that polycationic SMOF-1 adsorbed anionic aspartic acid-derived dipeptides (L,L and D,D) or tripeptide (L,L,L; Supplementary Fig. 15), as revealed for a previously reported SOF[42]. However, the solution of SMOF-1 in water did not exhibit induced circular dichroism signals within the range of 250–600 nm in the presence of an excess of these chiral peptides (Supplementary Fig. 16), indicating that no chirality bias was generated for SMOF-1.

On slow evaporation at ambient temperature, the solution of SMOF-1 slowly solidified and finally formed microcrystals, as evidenced by transmission electron microscope (TEM) with the selected area electron diffraction (SAED) and scanning electron microscope (SEM) images (Fig. 4a and Supplementary Fig. 17). The X-ray diffraction profile of the microcrystals exhibited three peaks centred at 3.0, 2.1 and 1.7 nm (Fig. 3d), respectively, which matched well with the {100}, {110} and {111} (1.7 nm) spacings of the modelled 3D framework. The SAXS profile displayed a strong, sharp peak at 3.0 nm (Fig. 3c), which was also observed on the 2D synchrotron X-ray scattering profile (Fig. 3g). Both peaks were related to the {100} spacing of the modelled framework. Cryo-TEM further revealed the periodicity of the microcrystals with the 3.0 and 2.1 nm lattice spacings (Fig. 4b,c and Supplementary Fig. 18), which again matched exactly with the simulated value for the lattice distance of {100} and {110} faces. These observations collectively provided consistent evidences to support that SMOF-1 maintained its periodicity in the solid state. Thermogravimetric analysis showed that the solid-state SMOF-1 was stable at a temperature above 300 °C (Supplementary Fig. 19).

On the basis of the X-ray diffraction and TEM results and the reported crystal parameters of the 1:2 complex of CB[8] and 4-phenyl-1-pyridinium[47], we could estimate the cubic unit-cell metrics of SMOF-1 as: a = b = c = 29.8 Å, α = β = γ = 90°. SAED patterns of the microcrystals with the reciprocal lattice observed for the {100} facet showed high foursquare order (Fig. 4a insert), which further confirmed the cubic unit cell of the microcrystals. The SAED pattern, that is, 1.5 nm for the {200} lattice spacing, was identical to the simulated data (1.5 nm). Elemental energy dispersive X-Ray spectroscopy (EDX) mapping analysis for the microcrystals also confirmed the compositions of the C, N, O, Ru and Cl elements (Supplementary Fig. 20).

**WD-POM encapsulation.** From the modelled 3D framework, we estimated the void volume of cubic SMOF-1 to be ~80%. The pore aperture, defined by the four CB[8] units in one self-assembled macrocycle adopting a square-like conformation, was calculated to be ca. 1.5 nm. The nitrogen gas absorption amount of SMOF-1 microcrystals at 77 K was determined to be 17.3 cm$^3$ (STP) g$^{-1}$ at P/P$_0$ = 1.0 (Supplementary Fig. 21), and the Brunauer–Emmett–Teller (BET) surface area of the microcrystals was estimated to be 36.4 m$^2$ g$^{-1}$, and the carbon dioxide gas absorption amount at 273 K was measured to be 3.86 cm$^3$ (STP) g$^{-1}$ at 800 mm Hg (Supplementary Fig. 22). These preliminary results showed that polycationic SMOF-1 exhibited only weak ability of adsorbing gas, as revealed for other polycationic frameworks[42]. As a new self-assembled cationic

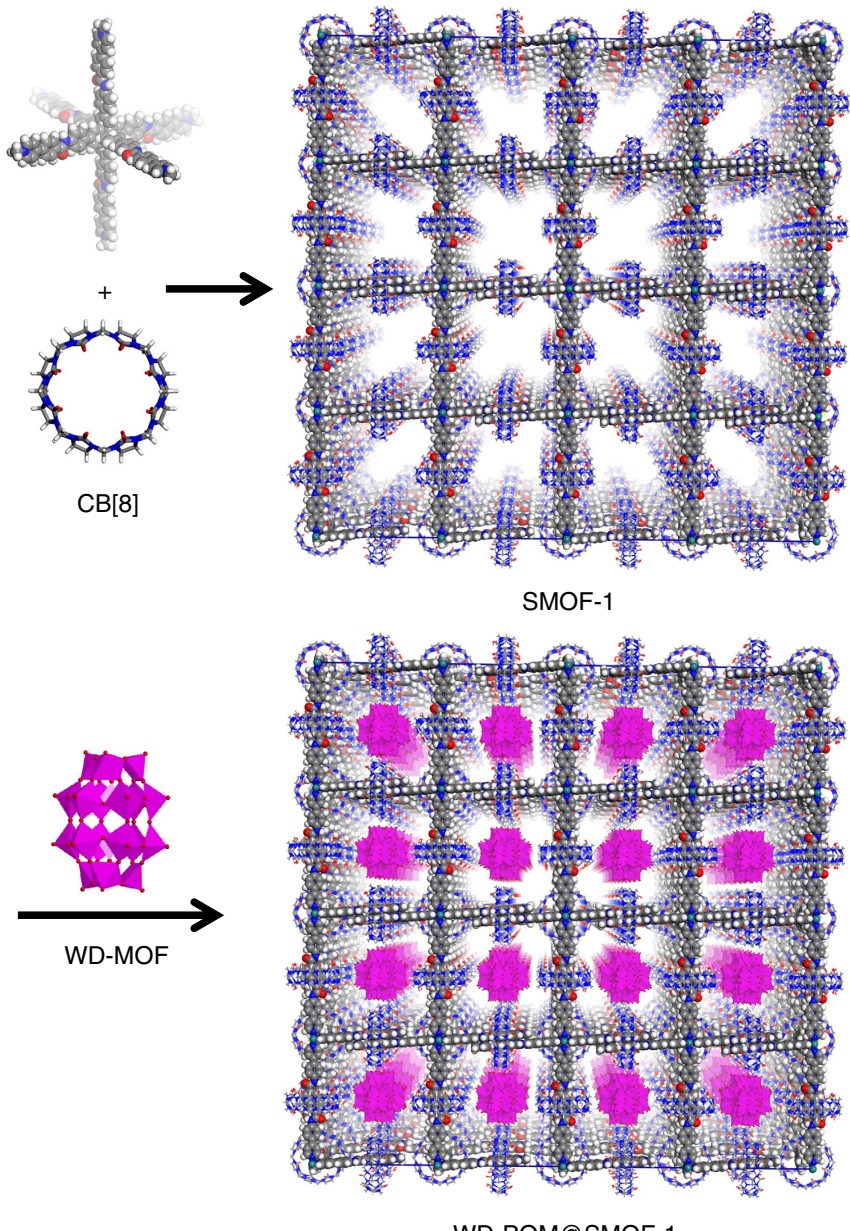

**Figure 2 | Self-assembly of 3D cubic SMOF-1 and WD-POM loading.** Formation of SMOF-1 and WD-POM@SMOF-1.The space-filling structural models were obtained using Materials Studio 7.0. H, white; C, light grey; N, blue; O, red; Ru, cyan; WD-POM ($[P_2W_{18}O_{62}]^{6-}$), purple polyhedron.

polyelectrolyte, SMOF-1 in water accommodated bulky functional anionic species, such as redox-active WD-POM $[P_2W_{18}O_{62}]^{6-}$, which has a width of about 1.1 nm. Such an exchange may be rationalized by the formation of soft acid ($[Ru(bpy)_3]^{2+}$)-soft base ($[P_2W_{18}O_{62}]^{6-}$) ion pairs and hard acid ($Na^+$)-hard base ($Cl^-$) ion pairs.

Adding WD-POM to the solution of SMOF-1 ($[\mathbf{1}] = 0.02$ mM) led to significant quenching of the fluorescence of the $Ru(bpy)_3^{2+}$ unit. Remarkably, addition of 1 equivalent of WD-POM (relative to [**1**]) could reach maximum quenching (Fig. 5a), and further addition of WD-POM did not affect the quenching anymore. This observation indicated that the adsorption of WD-POM by SMOF-1 occurred in a manner that one cubic cage of SMOF-1 encapsulated one WD-POM unit. This one-cage-one-guest formation was confirmed by the inductively coupled plasma-atomic emission spectrometry (ICP-AES) analysis, which revealed a Ru/W atomic ratio of 0.056 (calculated value:

0.056) for WD-POM@SMOF-1 after its aqueous solution ($[\mathbf{1}] = 3.0$ mM) was dialysed for 3 days in a bag with apertures of 1.5 nm diameter. This implied that every $Ru(bpy)_3^{2+}$ or encapsulated WD-POM cluster was mutually surrounded by eight counterparts at the vertices of a cubic cage. Such an encapsulation pattern is ideal for photo-initiated electron transfer from excited $Ru(bpy)_3^{2+}$ to redox-active WD-POM[49]. In contrast, quenching of the fluorescence of **1** by WD-POM in the absence of CB[8] under identical conditions was minimal.

The solution-phase synchrotron SAXS profile of the SMOF-1 solution containing 1 equivalent of WD-POM (relative to **1**) afforded a strong peak at 3.1 nm, which corresponded to the {100} peak (Fig. 3e). DLS experiments showed that, after the addition of WD-POM, the $D_H$ value of the solution of SMOF-1 ($[\mathbf{1}] = 3.0$ mM) in water was changed from 190 nm to 220 nm (Supplementary Figs 13 and 23). These observations indicated that SMOF-1 maintained its periodicity after adsorbing

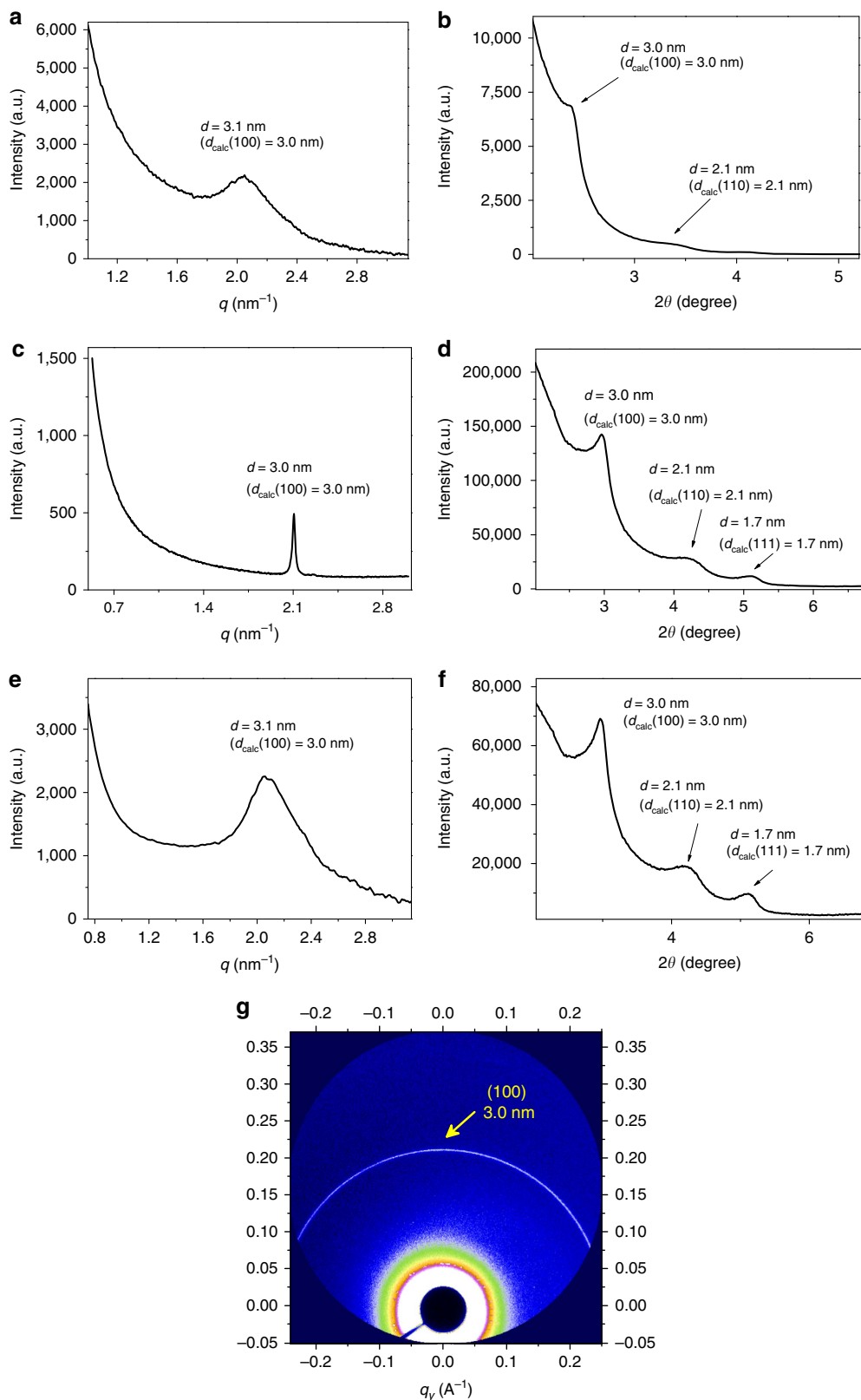

**Figure 3 | SAXS and X-ray diffraction profiles of 3D SMOF-1.** (**a**) Solution-phase synchrotron SAXS ([**1**] = 3.0 mM) in water. a.u., arbitrary unit. (**b**) Solution-phase synchrotron X-ray diffraction ([**1**] = 3.0 mM) in water. (**c**) Solid-phase SAXS. (**d**) Solid-phase X-ray diffraction. (**e**) Solution-phase synchrotron SAXS of the aqueous solution of WD-POM-encapsulated SMOF-1. [**1**] = 3.0 mM, [WD-POM] = 0.2 mM. (**f**) Solid-phase X-ray diffraction of WD-POM@SMOF-1. The sample was obtained by slow evaporation of the aqueous solution. [**1**] = 3.0 mM, [WD-POM] = 0.2 mM. (**g**) 2D solid-phase synchrotron X-ray scattering of SMOF-1. The peak values in **a**–**f** were attributed by choosing the position that was highest above the straight line defined by the two saddle points of the broad peak.

WD-POM in solution. Slow evaporation of the solvent led to the formation of microcrystals. The X-ray diffraction profile of the microcrystals exhibited three peaks at 3.0, 2.1, and 1.7 nm (Fig. 3f). These peaks respectively corresponded to the {100}, {110} or {111} spacing, supporting the periodicity of WD-POM-encapsulated SMOF-1 in the solid state. The peaks in Fig. 3e,f are all notably stronger than the corresponding ones of the **WD-POM**-free sample (Fig. 3a,d), suggesting an enhanced periodicity in the WD-POM-encapsulated SMOF-1. After the addition of WD-POM, the average crystallite size of SMOF-1 changed from 50 nm to 53 nm, which were estimated from the X-ray diffraction experiments (Fig. 3d,f) using the Debye–Scherrer equation. High-resolution TEM images revealed the 3.0 nm lattice spacing (Fig. 4d,e, and Supplementary Fig. 24) of WD-POM-encapsulated SMOF-1 from the {100} and {110} directions, which could be observed clearly under ambient temperature without using the cryo conditions required for the WD-POM-free SMOF-1 sample (Fig. 4b,c, and Supplementary Fig. 18). The high-angle annular dark field scanning TEM image also exhibited the 3.0 nm lattice spacing for the WD-POM-encapsulated microcrystals (Fig. 4f). Composition analysis by EDX mapping experiment indicated the presence of C, N, O, Ru, P, W and Cl elements (Supplementary Fig. 25), which was consistent with WD-POM encapsulation within SMOF-1. Taking altogether, these observations further confirmed the periodicity of WD-POM-encapsulated SMOF-1 in the solid state. The nitrogen gas absorption amount of WD-POM@SMOF-1 microcrystals at 77 K was determined to be 4.48 $cm^3$ (STP) $g^{-1}$ at $P/P_0 = 1.0$ (Supplementary Fig. 26), and the BET surface area of the microcrystals was estimated to be 7.19 $m^2 g^{-1}$.

**Photo-driven hydrogen production**. Efficient fluorescence quenching reflected that the electron transfer from the photo-excited state of the $[Ru(bpy)_3]^{2+}$ units to the encapsulated neighbouring WD-POM anions was remarkably enhanced through the unique one-cage-one-guest encapsulation pattern[49].

The highest occupied molecular orbital energy of complex **1** and the lowest unoccupied molecular orbital energy of WD-POM were determined to be $-3.59$ and $-4.78$ eV, respectively (Supplementary Table 3 and Supplementary Figs 27–29), also supporting the possibility of using WD-POM@SMOF-1 assemblies for catalysing visible light-driven proton reduction. A recent report illustrated that[49], under the irradiation of visible light at 450 nm, $[Ru(bpy)_3]^{2+}$-bearing MOFs enabled fast multi-electron injection from excited $[Ru(bpy)_3]^{2+}$ to encapsulated WD-POM, which in turn catalysed water-splitting half-reactions to produce $H_2$. In the present study, the formation of SMOF-1 caused the visible range absorption of **1** to shift from 470 to 500 nm (Supplementary Fig. 30), which overlaps better with the solar irradiance spectrum that peaks around 500 nm wavelength. We thus selected the 500 nm visible light as the excitation to investigate the efficiency of WD-POM@SMOF-1-catalysed proton reduction for $H_2$ production. The reactions were performed in an acidic aqueous solution (pH = 1.8) using methanol as the sacrificial electron donor[49,50].

We first investigated the $H_2$ production efficiency by keeping $[1] = 0.3$ mM and changing the concentration of encapsulated WD-POM from 0.002 to 0.6 mM. After irradiation for 12 h, it was found that, at (WD-POM) = 0.002 mM, which corresponded to a (**1**)/(WD-POM) ratio of 150, the turnover number (TON) for $H_2$ production reached 392 (defined as $n(1/2H_2)/n(WD-POM)$; (Fig. 5b), which corresponded to a $H_2$ evolving rate, that is, turnover frequency (TOF), of 3,553 $\mu mol\, g^{-1}\, h^{-1}$ (based on WD-POM). This level of $H_2$ production is about four times higher than that of a heterogeneous WD-POM@$[Ru(bpy)_3]^{2+}$-MOF system which bore the identical number of WD-POM and $[Ru(bpy)_3]^{2+}$ units[49]. The high efficiency of the WD-POM@SMOF-1 system may be attributed to its unique one-cage-one-guest encapsulation pattern and homogeneity, which not only allowed for quick diffusion and close contact of hydronium and methanol molecules, but also facilitated the electron transfer from excited $[Ru(bpy)_3]^{2+}$ to WD-POM. By keeping $[1]/[WD-POM] = 15$ and irradiation for 7 h, the $H_2$

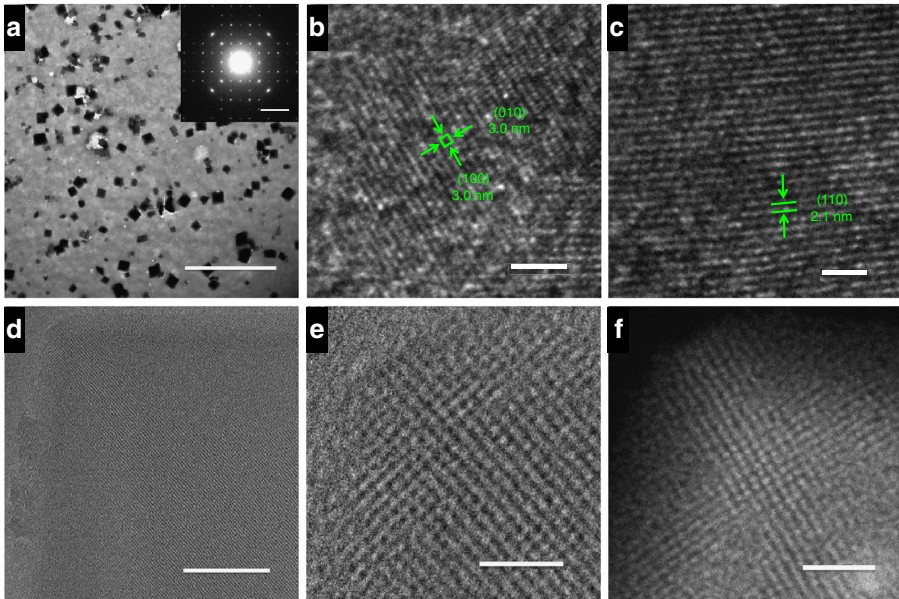

**Figure 4 | TEM images of the solid samples.** (**a**) TEM images of SMOF-1. Scale bar, 10 μm. Inset: SAED pattern showing the reciprocal lattice observed for the {100} facet, which showed foursquare order. Scale bar, 2 nm. (**b,c**) High-resolution cryo-TEM images of SMOF-1 from different facets showing different lattice spacings. (**b**) Scale bar, 20 nm. (**c**) Scale bar, 10 nm. (**d**) HR-TEM image of solid WD-POM@SMOF-1 from the {100} direction. The sample was obtained by slowly evaporating the aqueous solution ([**1**] = 0.3 mM, [WD-POM]/[**1**] = 0.067). Scale bar, 100 nm. (**e**) High-resolution TEM image of solid WD-POM@SMOF-1. Scale bar, 20 nm. (**f**) HAADF-STEM image of solid WD-POM@SMOF-1. Scale bar, 20 nm.

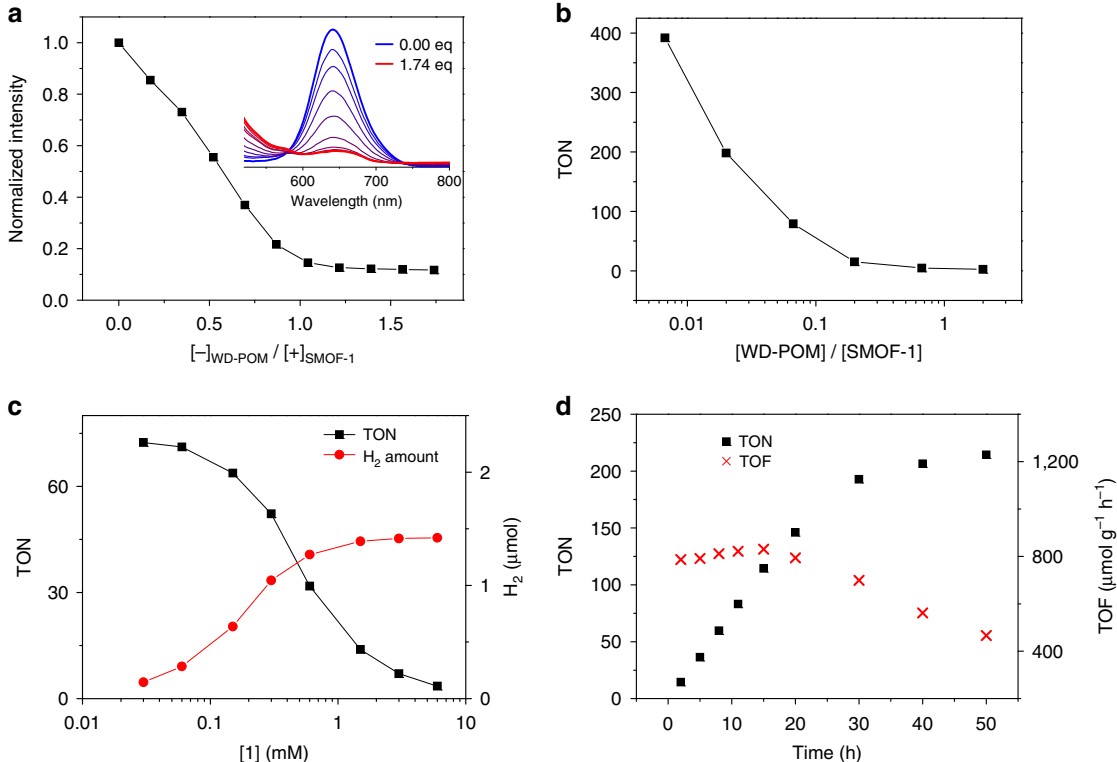

**Figure 5 | Fluorescence quenching and photo-driven hydrogen production. (a)** Normalized fluorescence ($\lambda = 640$ nm, $\lambda_{ex} = 500$ nm) quenching of SMOF-1 ([**1**] $= 0.02$ mM) by WD-POM in water. Inset: quenched fluorescence spectra. [$-$]/[$+$] $= 0$-1.74. [$-$] and [$+$] represent the total charge molar amount. **(b)** TONs of the solution of WD-POM@SMOF-1 ([**1**] $= 0.3$ mM) after irradiating for 12 h. [WD-POM]/[**1**] $= 0.0067$-2. **(c)** TONs of the solution of WD-POM@SMOF-1 and $H_2$ production amount after irradiating for 7 h. [**1**]/[WD-POM] $= 15$, [**1**] $= 0.03$ to 6 mM. **(d)** Time-dependent TON and TOF of the solution of WD-POM@SMOF-1. [**1**] $= 0.3$ mM, [WD-POM] $= 0.02$ mM. Methanol was used as the sacrificial electron donor.

production efficiency was explored at different hybrid concentrations ([**1**] = 0.003–6.0 mM). It was found that, with the increase of the amount of the hybrid, TON monotonically decreased due to continuously decreased excitation yield of the $[\text{Ru(bpy)}_3]^{2+}$ units as a result of the fixed irradiation capacity (Fig. 5c). However, the total $H_2$ output increased gradually and reached maximum at [**1**] = ca. 1 mM. Further increase of the hybrid did not obviously increase the $H_2$ output probably because of the decrease of the light transmittance. Longer irradiation was also implemented for the hybrid with [WD-POM] = 0.02 mM and [**1**] = 0.3 mM (Fig. 5d). After 50 h, TON reached 214. Nevertheless, the TOF was highest and remained unchanged within the first 20 h and then started to decrease, probably due to decomposition of the $[\text{Ru(bpy)}_3]^{2+}$ complex (Supplementary Fig. 31)[49]. After irradiation for 20 h, the $D_H$ value of the aqueous solution of WD-POM@SMOF-1 was not changed (Supplementary Fig. 23). Control experiments showed that irradiating the solution of SMOF-1, complex **1**, WD-POM, or the mixture of **1** and WD-POM for 10 h under the above conditions did not lead to $H_2$ production, confirming that the framework of SMOF-1 played a crucial role in catalysing the $H_2$ production by encapsulating WD-POM. The new WD-POM@SMOF-1 hybrid system could be recovered by evaporating the solvent under reduced pressure and was used for six times for hydrogen production. TON was reduced from 76 to 65 after being used for four times (Supplementary Table 1).

Heterogeneous photo-driven $H_2$ production was also investigated for WD-POM@SMOF-1 microcrystals (molar ratio: 1:15) as catalysts in acetonitrile and $N,N$-dimethylformamide (3:7) mixed medium (Supplementary Table 2), in which the microcrystals were insoluble. With methanol as the sacrificial electron donor, TON reached 48 after irradiation for 12 h. Using triethanolamine as the sacrificial electron donor, TON could reach 1,820 after irradiation for 14 h, which is ~5 times higher than that of the heterogeneous WD-POM@$[\text{Ru(bpy)}]_3^{2+}$-MOF hybrid[49]. When the amount of the loaded WD-POM was increased, TON decreased, but the rate of $H_2$ production was increased. After irradiation for 14 h, the average crystallite size of WD-POM@SMOF-1 changed from 53 to 60 nm (Supplementary Fig. 32). Ultraviolet-visible experiments showed that, for both the homogeneous and heterogeneous systems, WD-POM did not escape from SMOF-1 to the solution after irradiation for 50 h.

## Discussion

We have reported a self-assembly strategy for the generation of the first supramolecular MOF (SMOF-1) or metal-cored supramolecular organic framework in water at room temperature. SMOF-1 represents a new class of self-assembled frameworks that possess periodicity and porosity, the two key features of solid MOFs, yet **SMOF** offers more advantages over MOF as the former can exist in both solution and the solid state. Efficient encapsulation of WD-POM in SMOF-1 enables the creation of new efficient photo-catalysis system that exhibit a hydrogen production activity higher than that of the reported heterogeneous MOF-based system. The new self-assembly strategy alleviates the need for high temperature and long reaction time, and the formation of the supramolecular framework proceeds homogeneously. Thus, the work should open many possibilities for the construction of other functional porous materials in the future. For example, frameworks with enlarged apertures may be attained by using rigid ligands with systematically elongated

arms. Such new porous frameworks may be used to tune the photophysical and photochemical properties of functionalized nanoparticles through encapsulation. The mildness of the self-assembly strategy may also allow for the introduction of different functional groups and post-synthetic modifications for rationally designed frameworks, which would lead to new properties and applications.

## Methods

**Materials and measurements.** All reagents were obtained from commercial suppliers and used without further purification unless otherwise noted. All reactions were carried out under a dry nitrogen atmosphere. All solvents were dried before use following standard procedures. Column chromatography was performed on silica gel (200–300 mesh), and thin-layer chromatography (TLC) was performed on precoated silica gel plates (0.4–0.5 mm thick). $^1H$ and $^{13}C$ NMR spectra were recorded with a 400 MHz spectrometer in the indicated solvents at 25 °C (Supplementary Figs 33–42). $^1H$ NMR diffusion-ordered spectroscopic experiments were carried out with a 400 NMR spectrometer. Chemical shifts were referenced to the residual solvent peaks. Mass spectra (ESI) were obtained on Shimadzu LCMS-2010EV, IonSpec 4.7 Tesla FTMS and microTOF II spectrometers (Supplementary Fig. 43). DLS experiments were performed on a Malvern Zetasizer Nano ZS90 light scattering Instrument. Powder X-ray diffraction measurements were carried out on a Bruker D8 Advance diffractometer at 40 kV and 40 mA with Cu Kα radiation ($\lambda = 1.5406$ Å). Scanning electron micrographs and elemental distribution of the samples were obtained on a JSM-6330F Field Emission SEM combined with EDX analysis. Transmission electron micrographs were recorded on a 2100F JEOL FETEM microscope at 120 kv aligned for low dose ($10 \, e \, Å^{-2} s^{-1}$) diffractive imaging. Luminescence measurements were performed on a VARIAN CARY Eclipse Fluorescence Spectrophotometer. Ultraviolet-visible spectra were performed on a Perkin-Elmer 750 s instrument. Inductively coupled plasma-atomic emission spectroscopy (ICP-AES P-4010, Hitachi, Tokyo, Japan) was used to determine the Ru and W contents. The synthesis and characterization, and spectroscopic measurements, including DLS, thermogravimetric analysis, gas adsorption experiments, visible light-driven hydrogen production experiments and crystallite size calculations are summarized in the Supplementary Methods. The Spectra/Por 6 Dialysis Tubing (10 kDa Molecular Weight Cut Off, 8 mm Flat-width) was purchased from Spectrum Laboratories, Inc.

**Solution-phase synchrotron X-ray diffraction data collection.** The X-ray diffraction data were obtained at beamline BL14B1 of the Shanghai Synchrotron Radiation Facility (SSRF) at a wavelength of 1.2398 Å. BL14B1 is a beamline based on bending magnet and a Si (111) double-crystal monochromator was used to monochromatize the beam. The size of the focus spot is about 0.5 mm and the end station is equipped with a Huber 5021 diffractometer. NaI scintillation detector was used for data collection.

**Solid-phase synchrotron X-ray scattering data collection.** Synchrotron radiation SAXS experiments were performed on the BL16B beamline of Shanghai Synchrotron Radiation Facility (SSRF), using a fixed wavelength of 0.124 nm, a sample-to-detector distance of 1.85 m and an exposure time of 600 s. The 2D scattering pattern was collected on a charge coupled device camera, and the curve intensities versus q were obtained by integrating the data from the pattern.

**Solution-phase synchrotron SAXS data collection.** The data were collected at the SIBYLS Beamline 12.3.1 of the Advanced Light Source (ALS, Lawrence Berkeley National Laboratory). The SMOF-1 ([**1**] = 3.0 mM) complex in water were exposed for 0.5, 1 and 6 s, followed by a 1 s exposure to check for radiation damage, using a MARCCD X-ray detector system located 1.6 m from the sample chamber to collect data in the $q$-spacing 0.01–0.32 Å$^{-1}$, where $q = 4\pi\sin\theta/\lambda$ ($2\theta$ is the scattering angle and $\lambda$ is the wavelength). Filtrates resulting from the sample concentration process were exposed for equivalent times and the intensities were subtracted from the corresponding sample exposures. Data from the low- and high-resolution ranges of the respective short and long exposures were scaled and merged to obtain the final data sets.

**Data availability.** The authors declare that all relevant data are available from the authors on request.

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

## Acknowledgements

We acknowledge the Ministry of Science and Technology (2013CB834501), the Ministry of Education (Doctor Fellowship Grant), the National Natural Science Foundation (21,432,004, 21,529,201, 91,527,301) of China, the Science and Technology Commission of Shanghai Municipality (13NM1400200) for financial support and Shanghai Synchrotron Radiation Facility for providing BL16B beamline for collecting the synchrotron X-ray scattering and diffraction data. Y.L. thanks the support from the Molecular Foundry, Lawrence Berkeley National Laboratory, supported by the Office of Science, Office of Basic Energy Sciences, Scientific User Facilities Division, of the U.S. Department of Energy under the Contract No. DE-AC02-05CH11231.

## Author contributions

Z.-T.L., J.T., Y.L. and H.W. conceived and designed the research, analysed the data and composed the manuscript. J.T. performed most of the experiments. Z.-Y.X., S.-H.X., D.-W.X., Y.-H.R. and H.W. performed part of the experiments, D.-W.Z. analysed the data. All authors discussed and commented on the manuscript.

## Additional information

**Competing financial interests:** The authors declare no competing financial interests.

