## [Peer Review File · Nature Communications]

Reviewers' comments:

Reviewer #1 (Remarks to the Author):

The paper by Li et al represents a nice demonstration of this research groups strategy to make supramolecular materials. The paper builds upon their previous research (references 41, 42) in making materials through the use of CB macrocycles acting as linker holding different components together. In this instance they make a 3D MOF which can then act as a host for POMs. In turn this POM-MOF material can act as a catalyst for H₂ production. Overall the paper is an interesting example of a synthetic strategy which is then put to use for a real application (H₂ production) and in total I think the paper is a significant step beyond their previous research.

I have a few comments which the authors should address.

Page 4: The authors refer to Job's plots - this should read Job plots. The same is true in the SI.

Page 4: It is not clear what CP-a, CP-b and CP-c are. Could then authors put a figure, probably in the SI, to explain what they think these compounds actually are?

Page 5: The authors report DLS measurements for three materials, 2.7, 3.6 and 164nm. What is the error on these measurements. Are the author's justified in quoting values to this level of precision?

Page 6/7: The authors state that "we chose to simulate the 3D structure from only the Λ enantiomer, because parameters that define the periodicity of the two resulting frameworks should be identical." I agree with the statement but it does raise the question of whether both enantiomers are present in the material. Is there any chiral recognition in the formation of the materials? Have the authors recorded circular dichroism spectra?

Page 10: polyelectrolyte is spelt incorrectly.

Page 13: The authors comment that the material "can be used for at least three times by simply dissolving in water without obvious loss of the activity". Is three times the limit of what they have studied? What does "obvious loss" mean, is there any loss of activity and how was this measured?

Reviewer #2 (Remarks to the Author):

This manuscript reports a water-soluble supramolecular metal-organic framework (SMOF), the formation of which was induced by host-guest complexation between cucurbit[8]uril (CB[8]) and a hexaarmed ruthenium complex, and its application to photocatalytic hydrogen production. Extending their previous study on a water-soluble periodic supramolecular organic framework and encapsulation of anionic substances in the framework (Nature Communications, 2014), the authors prepared a hexaarmed ruthenium complex to achieve another 3D porous structure through the complexation of two aromatic arm moieties within CB[8]. The SMOF-1 was further utilized to absorb Well-Dawson-type polyoxometallates (WD-POMs) and the resulting WD-POM@SMOF-1 showed a photocatalytic activity for hydrogen production.

This is certainly an interesting piece of work; however, the general strategy for the formation of a water-soluble periodic supramolecular framework and encapsulation of anionic species in the framework has been already reported by the same groups (Nature Communications, 2014). In addition, photocatalytic hydrogen production from Well-Dawson-type polyoxometallates (WD-POMs) inside metal-organic frameworks has been studied by the Lin group (Ref. 49). Therefore, conceptually, this work is not novel enough to be published in Nature Communications. Therefore, although interesting, I cannot recommend the publication of this work in Nature Communications. It may be more suitable for Journal of American Chemical Society or other journals. A couple of comments/suggestions are as follows:

1. As the authors mentioned, TON and TOF of WD-POM@SMOF-1 is quite high at the very low loading of WD-POM. However, the total production of H₂ is quite little (less than 1.5 micro mole even after 7 hours reaction), compared to other MOF systems (Energy Environ. Sci., 2013, 6,

3229, Chem. Commun., 2014, 50, 6779). In terms of H₂ production, the authors should discuss the merits and demerits of their system in comparison with existing systems.

2. The authors alluded the decrease in TOF after 20 hours to the decomposition of the Ru complex. I suggest the authors to check whether there is any leaching of WD-POM and/or decomposition of SMOF-1 during the H₂ production.

Reviewer #3 (Remarks to the Author):

The present paper deals with the development of unique water soluble supramolecular metal organic frameworks and their applications for photocatalytic H₂ production reaction from liquid phase involving various kinds of sacrificial reagents. Since this is the first development of water soluble supramolecular metal organic frameworks having photocatalytic activity, to the best of my knowledge, I feel this paper can be accepted after the proper considerations on the following points.

1. Authors insist the formation of one-cage-one-guest formation for POM@SMOF-1, however, one-cage-one-guest formation should be clarified by the chemical composition analysis such as CHN analysis and ICP measurements.

2. The BET surface area of SMOF-1 is very small (36.4m²g⁻¹), however, it can accommodate the large number of POM in the pore. Is this gap between the surface area and anion exchange capacity reasonable when the theoretical modeling is taken into the consideration? Please add some explanation on this points in the manuscript. Authors should provide the BET surface area of SMOF-1 after accommodation of POM in the pore (POM@SMOF-1).

3. Please estimate and provide the primary particle size of SMOF-1 and POM@SMOF-1 in water and solid state, respectively, by XRD diffraction data or other spectroscopic data and compare with those after the photocatalytic reaction.

4. Please provide the reason why photocatalytic reactions were carried out in the acidic condition. What happens when reactions are performed at higher pH region?

5. What is the source of H₂ evolved from methanol dissolved in acetonitrile or N,N-dimethylformamide? If the H₂ is originated from methanol, what is the reaction product?

6. Please provide the estimated HOMO-LUMO level of Ru complex and POM to ensure the electron transfer from Ru complex to POM.

7. Normally, single crystal XRD measurements are indispensable to determine the detailed structure of MOF. Please provide the reason why single crystal XRD measurements are not provided in the manuscript.

Point-to-point response to the referees' comments:

(the position of the revisions represents that in the revised manuscript):

Reviewer #1 (Remarks to the Author):

- 1) Page 4: The authors refer to Job's plots - this should read Job plots. The same is true in the SI.

Response:

We have changed “Job’s plots” as “Job plots” in the main text and the Supplementary Information.

- 2) Page 4: It is not clear what CP-a, CP-b and CP-c are. Could then authors put a figure, probably in the SI, to explain what they think these compounds actually are?

Response:

(a) The proposed models of **CP-a** and **CP-b** have been included in Supporting Information as Supplementary Figure 7. Accordingly, supplementary figures following this had been re-numbered.

(b) In the text (pages 3-4): “Supplementary Figure 7” has been added after “**CP-a**” and “**CP-b**”.

(c) In the text (page 5, line 5 from bottom): “**CP-c**” was related to “**SMOF-1**” in this sentence: *“Both experiments supported that **CP-c** formed soluble periodic supramolecular metal-organic framework (**SMOF-1**) in water.”*

(d) The model of **CP-c** is the same as that of **SMOF-1** showed in Figure 2.

- 3) Page 5: The authors report DLS measurements for three materials, 2.7, 3.6 and 164nm. What is the error on these measurements. Are the author's justified in quoting values to this level of precision?

Response:

(a) The original values (2.70 ± 0.57 , 3.62 ± 0.87 , and 164 ± 32 , respectively) were afforded automatically by the instrument. Given the difference of the three values, we modified the values with three or two significance digits.

(b) For keeping consistency, in the text (page 4, line 3 from bottom), we have changed “2.7” and “3.6” as “2.70” and “3.62”, respectively.

(c) In the legend of Supplementary Figure 12, this description has been added: *“The peak values of the solutions of compounds **1-3** and **CB[8]** were originally afforded by the instrument as 164 ± 32 , 3.62 ± 0.87 and 2.70 ± 0.57 nm, respectively.”*

(d) Accordingly, the supplementary figures following this have been re-numbered.

- 4) Page 6/7: The authors state that "we chose to simulate the 3D structure from only the Λ enantiomer, because parameters that define the periodicity of the two resulting frameworks should be identical." I agree with the statement but it does raise the question of whether both enantiomers are present in the material. Is there any chiral recognition in the formation of the materials? Have the authors recorded circular dichroism spectra?

Response:

- (a) The circular dichroism spectra of **SMOF-1** ($[1] = 0.2$ mM) and complex **1** (0.2 mM) were recorded. Both samples exhibited no observable CD signals (250-600 nm).
- (b) To exploit possible chiral recognition, we first recorded the fluorescent spectra of **SMOF-1** in the presence of aspartic acid-derived anionic chiral dipeptides **asp-asp** (L,L and D,D) or tripeptide **asp-asp-asp** (D,D,D) and found that the fluorescence of **SMOF-1** was quenched remarkably by these anionic peptides. The result was consistent with that reported for a 3D polycationic SOF (Tian, J. et al. *Nat. Commun.* **2014**, *5*, 5574) and supported that the peptides were adsorbed by the porous **SMOF-1**. We then recorded the CD spectra of **SMOF-1** in water in the presence of the three chiral compounds. All the spectra did not exhibit observable induced CD signals (250-600 nm), indicating that the chiral molecules did not induce chirality bias for the 3D framework.
- (c) The corresponding fluorescent and circular dichroism spectra have been provided as Supplementary Figures 15 and 16. Figures following these figures have been re-numbered.
- (d) In the text (page 5, by the end of the last paragraph), this description has been included: *"Fluorescent experiments showed that polycationic **SMOF-1** adsorbed anionic aspartic acid-derived dipeptides (L,L and D,D) or tripeptide (L,L,L) (Supplementary Fig. 15), as revealed for a previously reported SOF.⁴² However, the solution of **SMOF-1** in water did not exhibit induced circular dichroism signals within the range of 250-600 nm in the presence of an excess of these chiral peptides (Supplementary Fig. 16), indicating that no chirality bias was generated for **SMOF-1**."*

- 5) Page 10: polyelectrolyte is spelt incorrectly.

Response:

Page 7, line 12: This has been corrected.

- 6) Page 13: The authors comment that the material "can be used for at least three times by simply dissolving in water without obvious loss of the activity". Is three times

the limit of what they have studied? What does "obvious loss" mean, is there any loss of activity and how was this measured?

Response:

- (a) We repeated the hydrogen production experiment for six times, the data of the 4th to 6th runs have been included in Supplementary Table 1.
- (b) Page 10, line 1 from bottom: This description has been changed as: “*The new WD-POM@SMOF-1 hybrid system could be recovered by evaporating the solvent under reduced pressure and was used for six times for hydrogen production. TON was reduced from 76 to 65 after being used for four times (Supplementary Table 1).*”

Reviewer #2 (Remarks to the Author):

- 1) As the authors mentioned, TON and TOF of WD-POM@SMOF-1 is quite high at the very low loading of WD-POM. However, the total production of H₂ is quite little (less than 1.5 micro mole even after 7 hours reaction), compared to other MOF systems (Energy Environ. Sci., 2013, 6, 3229, Chem. Commun., 2014, 50, 6779). In terms of H₂ production, the authors should discuss the merits and demerits of their system in comparison with existing systems.

Response:

- (a) The total production of H₂ by the two reported systems was about 700 μmol (MOF-253-Pt system, irradiating time: 6.5 h, [Pt] = 0.53 mM in 100 mL, Energy Environ. Sci., 2013, 6, 3229) and 10.5 μmol (Ti-MOF-Ru(tpy)₂ system, irradiating time: 7 h, [Pt] = 0.05 mM in 2 mL, Chem. Commun., 2014, 50, 6779). For our homogeneous and heterogeneous systems, the total H₂ production was 1.04 μmol ([WD-POM] = 0.02 mM, irradiating time: 7 h, 2.2 mL) and 3.64 μmol ([WD-POM] = 0.002 mM, irradiating time: 14 h, 2.0 mL). For comparison, we transferred these values to TOFs (turnover frequency: TON per hour). TOF was 4 h⁻¹ for MOF-253-Pt system, 35 h⁻¹ for Ti-MOF-Ru(tpy)₂ system, and 7 h⁻¹ and 130 h⁻¹ for our homogeneous and heterogeneous systems (Supplementary Table 2).
 - (b) Considering that these catalysts and their molar amount used, reaction times and solvents were all different, the above results cannot be used for reasonably assessing their merits and demerits. Thus, we did not revise the manuscript by including these results.
- 2) The authors alluded the decrease in TOF after 20 hours to the decomposition of the Ru complex. I suggest the authors to check whether there is any leaching of WD-POM and/or decomposition of SMOF-1 during the H₂ production.

Response:

- (a) Time-dependent experiments of the solution of **WD-POM@SMOF-1** were performed in a sealed glass bottle for the homogenous system. Thus, the leaching of **WD-POM** can be excluded.
- (b) The UV-vis spectrum of **WD-POM@SMOF-1** before and after irradiation showed that the Ru²⁺ complex was partially decomposed. The similar result was observed for a previously reported MOF-based catalyst (Lin, W. et al. *J. Am. Chem. Soc.*, 2015, 137, 3197-3200).
- (c) The UV-vis spectra have been provided as Supplementary Figure 31. Accordingly, supplementary figures following this have been re-numbered.
- (d) In the text (page 10, line 6 from bottom): “(Supplementary Figure 31)⁴⁹” has been added.

Reviewer #3 (Remarks to the Author):

- 1) Authors insist the formation of one-cage-one-guest formation for POM@SMOF-1, however, one-cage-one-guest formation should be clarified by the chemical composition analysis such as CHN analysis and ICP measurements.

Response:

- (a) Because the Wells–Dawson-type polyoxometalate contains no CHN, the cage-guest formation could not be clarified by the chemical composition analysis.
 - (b) ICP measurement was performed, which supported the one-cage-one-guest formation.
 - (c) In the text (page 7, line 3 from bottom): This description has been added: “*This one-cage-one-guest formation was confirmed by the inductively coupled plasma atomic emission spectrometry (ICP-AES) analysis, which revealed a Ru/W atomic ratio of 0.056 (calculated value: 0.056) for **WD-POM@SMOF-1** after its aqueous solution ([I] = 3.0 mM) was dialyzed for 3 days in a bag with apertures of 1.5 nm diameter.*”
 - (d) In the text (page 12, line 2 from bottom): The instrument and method have been added to the end of the **Materials and measurements** section.
- 2) The BET surface area of SMOF-1 is very small (36.4m²g⁻¹), however, it can accommodate the large number of POM in the pore. Is this gap between the surface area and anion exchange capacity reasonable when the theoretical modeling is taken into the consideration? Please add some explanation on this points in the manuscript. Authors should provide the BET surface area of SMOF-1 after accommodation of POM in the pore (POM@SMOF-1).

Response:

- (a) Different from the adsorption of gas on the surface of MOFs or COFs, the accommodation of anionic POM into the pores of cationic **SMOF-1** was a process of ion exchange, which involved the formation of ion pairs of soft acid (Ru^{2+} complex) and soft base ($[\text{P}_2\text{W}_{18}\text{O}_{62}]^{6-}$) and ion pairs of hard acid (Na^+) and hard base (Cl^-).
- (b) In the text (page 7, line 11): The related description has been changed as: “*As a new self-assembled cationic polyelectrolyte, **SMOF-1** in water accommodated bulky functional anionic species, such as redox-active Wells–Dawson-type polyoxoanions $[\text{P}_2\text{W}_{18}\text{O}_{62}]^{6-}$ (**WD-POM**), which has a width of about 1.1 nm. Such an exchange may be rationalized by the formation of soft acid ($[\text{Ru}(\text{bpy})_3]^{2+}$)-soft base ($[\text{P}_2\text{W}_{18}\text{O}_{62}]^{6-}$) ion pairs and hard acid (Na^+)-hard base (Cl^-) ion pairs.*”
- (c) In the text (page 9, line 4): This description has been added: “*The nitrogen gas absorption amount of **WD-POM@SMOF-1** microcrystals at 77 K was determined to be 4.48 cm^3 (STP) g^{-1} at $P/P_0 = 1.0$ (Supplementary Fig. 26), and the BET surface area of the microcrystals was estimated to be $7.19 \text{ m}^2 \text{ g}^{-1}$.*”
- (d) Nitrogen gas adsorption isotherm curves of the **WD-POM@SMOF-1** solid has been included in Supplementary Figure 26. Accordingly, figures following this has been re-numbered.
- 3) Please estimate and provide the primary particle size of SMOF-1 and POM@SMOF-1 in water and solid state, respectively, by XRD diffraction data or other spectroscopic data and compare with those after the photocatalytic reaction.

Response:

- (a) In the text (page 8, line 8): This description has been added: “*DLS experiments showed that, after the addition of **WD-POM**, the D_H value of the solution of **SMOF-1** ($[I] = 3.0 \text{ mM}$) in water was changed from 190 nm to 220 nm (Supplementary Fig. 13 and 23).*”
- (b) In the text (page 10, line 6 from bottom): This description has been added: “*After irradiation for 20 hours, the D_H value of the aqueous solution of **WD-POM@SMOF-1** was not changed (Supplementary Fig. 23).*”
- (c) In the text (Page 8, line 9 from the bottom): The related description has been changed as: “*After the addition of **WD-POM**, the average crystallite size of **SMOF-1** changed from 50 nm to 53 nm, which were estimated from the XRD experiments (Figures 3d and 3f) using the Debye-Scherrer equation.*”
- (d) In the text (page 11, line 11): This description has been added: “*After irradiation for 14 hours, the average crystallite size of **WD-POM@SMOF-1** changed from 53 nm to 60 nm (Supplementary Fig. 32).*”
- (e) The solid-phase XRD profile of **WD-POM@SMOF-1** after irradiation has been included as Supplementary Figure 32.
- (f) Accordingly, supplementary figures following this had been re-numbered.
- (g) The calculation method has been included in SI (page 40).

- 4) Please provide the reason why photocatalytic reactions were carried out in the acidic condition. What happens when reactions are performed at higher pH region?

Response:

- (a) This is a well-established method (see: *Energy Environ. Sci.* 2013, 6, 1504; *Chem. Rev.* 1998, 98, 219). In an acidic condition, multi-electron processes are facilitated to a large extent.
 - (b) With pH being increased from 1.8 to 4.2 and to 7.0, TON decreased from 76 to 7 and to 2, reflecting the decrease of the catalysis efficiency with the increase of pH.
 - (c) The results have been included in Supplementary Table 1.
 - (d) In the text (page 9, line 5 from bottom): We have added references (ref. 49 and 50) to indicate the acidic condition.
- 5) What is the source of H₂ evolved from methanol dissolved in acetonitrile or N,N-dimethylformamide? If the H₂ is originated from methanol, what is the reaction product?

Response:

- (a) The source of H₂ was methanol and the reaction product was formaldehyde. The mechanism had been reported (see: A. Ioannidis et al, *Inorg. Chem.* 1985, 24, 439-441).
 - (b) Since this is a previously established process, we did not conduct new experiments to support the mechanism.
- 6) Please provide the estimated HOMO-LUMO level of Ru complex and POM to ensure the electron transfer from Ru complex to POM.

Response:

- (a) In the text (page 9, line 11): This description has been added: “*The highest occupied molecular orbital (HOMO) energy of complex 1 and the lowest unoccupied molecular orbital (LUMO) energy of **WD-POM** were determined to be -3.59 eV and -4.78 eV, respectively (Supplementary Table 3 and Fig. 27-29), also supporting the possibility of using **WD-POM@SMOF-1** assemblies for catalyzing visible light-driven proton reduction.*”
- (b) Estimated molecular orbital energies of complex 1 and **WD-POM** have been included in Supplementary Table 3.
- (c) UV-vis spectra of complex 1 and **WD-POM** have been included in Supplementary Figure 27.

- (d) Cyclic voltammetry of **WD-POM** has been included in Supplementary Figure 28.
 - (e) Energy level diagrams for complex **1** and WD-POM have been included in Supplementary Figure 29.
 - (f) Accordingly, supplementary figures following this had been re-numbered.
- 7) Normally, single crystal XRD measurements are indispensable to determine the detailed structure of MOF. Please provide the reason why single crystal XRD measurements are not provided in the manuscript.

Response:

The effort for single crystal XRD measurements has not been successful at the current stage for our self-assembled system. Microcrystals had been obtained, as observed with TEM, but single crystals with the suitable size for measuring were not available. The large pores might contain a large number of water molecules and the CB[8]⊂(PhPy)₂ units might rotate, which were expected to reduce the crystallinity.

Reviewers' comments:

Reviewer #1 (Remarks to the Author):

The authors have addressed all the comments raised by the referees. I conclude that the paper is now suitable for publication in Nature Communications.

Reviewer #2 (Remarks to the Author):

The revised manuscript reports a water-soluble supramolecular metal-organic framework (SMOF) utilizing host-guest complexation between cucurbit[8]uril (CB[8]) and a hexaarmed ruthenium complex, and demonstrates its application to photocatalytic hydrogen production. The work has been improved by providing additional data, which were mostly requested by reviewer #3. However, the comment that raised by me was not addressed properly.

I suggested the authors to check leaching of WD-POM to make sure whether WD-POM escapes from WD-POM@SMOF-1 during the H₂ production or not. However, they seem to have misunderstood this comment.

Overall, I still stand by my previous position. Without any doubt this is a well-executed work, but I do not think it is novel enough to be published in Nature Communications as the concepts have already been demonstrated by the same and other groups as I pointed out earlier.

Reviewer #3 (Remarks to the Author):

The authors have addressed all the comments raised by the reviewers and I am satisfied with their revision.

Point-by-point response to the reviewers' comment:

(The position of the revisions represents that in the revised manuscript):

Reviewer #2 (Remarks to the Author):

Comment: I suggested the authors to check leaching of WD-POM to make sure whether WD-POM escapes from WD-POM@SMOF-1 during the H₂ production or not. However, they seem to have misunderstood this comment.

Response:

- (a) To address this issue, we have performed the UV-vis experiments for the solution and solid samples to detect if WD-POM escaped from WD-POM@SMOF-1 during the H₂ production. The results showed that no leaching of WD-POM occurred for both systems after irradiation for 50 hours.
- (b) In the text (page 11, the end of paragraph 2), we have added this description: “UV-vis experiments showed that, for both the homogeneous and heterogeneous systems, **WD-POM** did not escape from **SMOF-1** to the solution after irradiation for 50 hours.”
- (c) In the **Materials and measurements** section (page 13, the end of the first paragraph): we have added this description: “The Spectra/Por 6 Dialysis Tubing (10 kDa Molecular Weight Cut Off, 8 mm Flat-width) was purchased from Spectrum Laboratories, Inc.”
- (d) In SI (page 40): The procedures for the UV-vis experiments have been added.
- (e) In SI (Figure 30): The UV-vis spectra of **WD-POM** in water/MeOH and DMF/MeCN/triethanolamine/water have been included as Figures 30b and 30c.